# An Explainable Spatial-Temporal Graphical Convolutional Network to Score Freezing of Gait in Parkinsonian Patients

**DOI:** 10.3390/s23041766

**Published:** 2023-02-04

**Authors:** Hyeokhyen Kwon, Gari D. Clifford, Imari Genias, Doug Bernhard, Christine D. Esper, Stewart A. Factor, J. Lucas McKay

**Affiliations:** 1Department of Biomedical Informatics, School of Medicine, Emory University, Atlanta, GA 30322, USA; 2Department of Epidemiology, Rollins School of Public Health, Emory University, Atlanta, GA 30322, USA; 3Jean and Paul Amos Parkinson’s Disease and Movement Disorders Program, Department of Neurology, School of Medicine, Emory University, Atlanta, GA 30322, USA

**Keywords:** deep learning, motion capture, multi-task learning, Parkinson’s disease

## Abstract

Freezing of gait (FOG) is a poorly understood heterogeneous gait disorder seen in patients with parkinsonism which contributes to significant morbidity and social isolation. FOG is currently measured with scales that are typically performed by movement disorders specialists (i.e., MDS-UPDRS), or through patient completed questionnaires (N-FOG-Q) both of which are inadequate in addressing the heterogeneous nature of the disorder and are unsuitable for use in clinical trials The purpose of this study was to devise a method to measure FOG objectively, hence improving our ability to identify it and accurately evaluate new therapies. A major innovation of our study is that it is the first study of its kind that uses the largest sample size (>30 h, N = 57) in order to apply explainable, multi-task deep learning models for quantifying FOG over the course of the medication cycle and at varying levels of parkinsonism severity. We trained interpretable deep learning models with multi-task learning to simultaneously score FOG (cross-validated F1 score 97.6%), identify medication state (OFF vs. ON levodopa; cross-validated F1 score 96.8%), and measure total PD severity (MDS-UPDRS-III score prediction error ≤ 2.7 points) using kinematic data of a well-characterized sample of N = 57 patients during levodopa challenge tests. The proposed model was able to *explain* how kinematic movements are associated with each FOG severity level that were highly consistent with the features, in which movement disorders specialists are trained to identify as characteristics of freezing. Overall, we demonstrate that deep learning models’ capability to capture complex movement patterns in kinematic data can automatically and objectively score FOG with high accuracy. These models have the potential to discover novel kinematic biomarkers for FOG that can be used for hypothesis generation and potentially as clinical trial outcome measures.

## 1. Introduction

Parkinson’s Disease (PD) is a slowly progressive neurodegenerative disorder that predominantly affects dopamine-producing neurons in the brain, and individuals with PD exceed more than 10 million people worldwide [1,2]. One of the most disabling features of PD and one of the greatest unmet needs is freezing of gait (FOG), which unfortunately is not always clearly treatable medically and/or surgically. FOG is described as brief arrests of stepping when initiating gait, turning, or walking straight ahead [3,4,5]. When a person freezes, they feel like their feet are “glued” to the floor. FOG is a frequent cause of falls and serious injuries, and represents a significant public health burden (∼86% of patients fall each year) [6,7,8].

One critical factor limiting our ability to treat FOG is that clinicians measure it relatively coarsely, primarily with expert rater observations as part of the Movement Disorder Society-Unified Parkinson’s Disease Rating Scale Part III (MDS-UPDRS-III) scale [9]. This scale requires specially trained raters who have typically completed movement disorders training. In addition, despite being resource-intensive, FOG is only quantified with a single item on an ordinal scale from 0 to 4, which may be too insensitive to detect small beneficial effects. The most established self-reporting scale used in research settings, the N-FOG-Q is acknowledged to be insufficiently sensitive for clinical trial use [10]. Previous work have shown that FOG may be associated with non-dopaminergic system changes [3,11,12], which suggests the potential for new treatments beyond dopaminergic medications like carbidopa-levodopa [4]. However, developing a novel drug that is effective in treating FOG requires accurately quantifying FOG to increase the precision for clinical trials.

Multiple studies have proposed methods to phenotype and rate FOG from kinematic data during walking. For example, those include capturing impaired gait patterns from lower back motion [13], describing gait complexity as a topological nonlinear dynamics system [14], or exploring combinations of sensor locations (shank, thigh, waist), axes (orthogonal, mediolateral, and antero-posterior), window lengths, and features (statistical, frequency, and time-series) to find the best setting that captures FOG characteristics.

Much of the prior work is characterized by a few substantial limitations [15], including (1) a small number of body-worn sensor locations, (2) small sample sizes with mostly early-stage PD patients lacking of severe FOG cases, (3) little consensus on proposed methods across studies, and (4) a relative paucity of studies conducted in the ON- and OFF- medication states, which is necessary to develop technology that will work over the entire medication cycle.

More importantly, most prior studies rely on hand-crafted features for identifying FOG, which may neglect important latent features within the data. For example, relative power in a “freeze band” of accelerometry or other signals [16,17,18], peak detection or similar methods applied to body segment motion [19,20], cycle-to-cycle variation in gait parameters [21], or a combination of the above were used in a support vector machine or other shallow machine learning models [22]. Due to the variability and complexity of FOG behavior, it is unlikely that manually designed spectral features will capture all the characteristics of FOG phenotypes. The popular “freeze band” analysis cannot capture pure akinetic freezing, which does not present with tremulousness.

Here, we use a deep learning approach to capture complex patterns in kinematic data and automatically score FOG, as well as identifying medication state and measure total MDS-UPDRS-III score during a rigorous levodopa challenge paradigm [3]. The proposed explainable (or interpretable) deep learning model automatically identifies the most relevant body joint or limb part and motion segment that is useful while performing multi-task predictions. According to the proposed model, fine-grained movement differences are associated with varying FOG scores, showing the heterogeneity of FOG, in agreement with movement disorder specialists’ observations. Our findings could be used to create new clinical trials to discover new kinematic biomarkers or to develop fine-grained and objective severity scales for individualized FOG treatment. We analyzed over 30 h of 3D motion capture data of 57 patients with varying PD disease duration and FOG severity, including 5 patients with primary progressive FOG, a distinct condition in which FOG presents without parkinsonian features [5]. This dataset is among the largest samples seen in the FOG literature (in which the average sample size was recently estimated as 18±15 [15]). To our knowledge, this work is the first application of interpretable deep learning to solve such a multi-task problem in PD.

## 2. Materials Furthermore, Methods

We trained an interpretable deep learning model on whole-body 3D kinematic data taken from behavioral motor tasks in N = 57 patients with and without FOG. Clinical, imaging, and cerebrospinal fluid analysis results from patients in this sample have been reported previously [3,23].

### 2.1. Behavioral Testing

#### 2.1.1. Study Participants

Although this was an observational study for which registration was not required, it was registered through clinicaltrials.gov (NCT02387281). Participants were recruited from the Emory Movement Disorders Clinic and provided written informed consent according to procedures approved by Emory University IRB. The inclusion criteria included: Age ≥ 18 years; PD diagnosis according to United Kingdom Brain Bank criteria [24]; Hoehn & Yahr stage I-IV in the OFF state; ability to sign a consent document and willing to participate in all aspects of the study. Participants with FOG were additionally required to have FOG noted in medical history and confirmed visually by examiner. Exclusion criteria included: vascular parkinsonism and drug-induced parkinsonism as well as the presence of cerebrovascular disease or extensive white matter disease; prior treatment with medications that cause parkinsonism; neurological or orthopedic disorders interfering with gait; dementia or other medical problems precluding completion of the study protocol. Demographic and clinical characteristics of study participants are presented in Table 1.

#### 2.1.2. Levodopa Challenge Paradigm

Each participant was assessed twice using an identical testing protocol: first, in the practically defined “OFF” state > 12 h after the last intake of all antiparkinsonian medications, and second, after a levodopa equivalent dose of ∼150% of the typical morning dose sufficient to elicit a full “ON” state. Additional details of the levodopa testing procedure have been presented previously [3]. In each state, they were assessed with the MDS-UPDRS-III motor exam [9] and with timed-up-and-go (TUG) tests in the motion capture laboratory [25] in normal and cognitive dual-task conditions [26], with three replicates each. Patients were instructed to turn left on all TUG tests, consistent with our clinical testing paradigm. Performance was scored in person, and scores were confirmed from video if necessary.

#### 2.1.3. Motion Capture

TUG tests were recorded using 3D optical motion capture (Motion Analysis Corporation, Santa Rosa, CA, USA). The motion capture facility is located in our clinical center and measures 5.8 m × 9.0 m with a capture area of 3.0 m × 4.6 m, and is equipped with 14 Osprey cameras with a resolution of 640 × 480 running at 120 Hz. During the testing session, patients wore tight-fitting clothes and were instrumented with reflective adhesive markers as recommended by the motion capture system manufacturer, configured as a superset of the Helen-Hayes kinematic marker set [27], incorporating additional markers on the hands. An example of the kinematic marker data is shown in Figure 1. Prior to analysis, all kinematic data were projected to a hip-centered coordinate system and normalized to zero mean and unit standard deviation.

### 2.2. Modeling

#### 2.2.1. Model Overview

Applications for deep learning models range from industry [28] to healthcare [29], enabling breakthroughs in complex data analysis. Those also include human activity or motion analysis using a kinematic time-series dataset [30,31]. While high recognition accuracy is important, understanding the reasoning behind the model’s decisions is also essential. To understand these black-box models, various explainable deep learning techniques are introduced to detect complex patterns in time-series data to *explain* the model behavior in a human-interpretable manner [32]. This work uses a kind of interpretable deep learning methods, namely, attention-based approach [33,34], to quantify FOG in parkinsonism.

Our proposed model is an attention-based adaptive graphical convolutional network (AGCN, [34]) with adaptive trimming [35]. The overall model architecture is shown in Figure 2. We process the 3D motion capture data following a common deep learning-based human activity recognition paradigm [30,36]. Motion capture data from each testing sequence is comprised of three channels (x, anterior/posterior; y, lateral; z, vertical) for each of 60 kinematic markers, for a total of 180 independent channels. The data from each sequence is segmented into analysis windows of 4 s, N×C×T, where N=60, C=3, and T=480 for 120 Hz signals, with 1 s intervals. A 4 s analysis window is chosen to capture a sufficient duration of FOG episodes while patients are walking [37].

Each 4 s analysis window is labeled with medication state (OFF/ON), FOG score (0, 1, 2, 3, or 4, from MDS-UPDRS-III item 3.11), and MDS-UPDRS-III total score, excluding the FOG score. The proposed model is trained to predict the labels for each analysis window based on the 3D kinematic data.

For extracting kinematic features from each 4 s window, the proposed model considers two aspects: (1) the core motion segment, which corresponds to the most relevant section of time within each window, and (2) the most relevant kinematic marker (joint) and edge between markers (bone) for the given multiple prediction tasks. The model uses adaptive trimming (AT) to identify the core motion segment within each 4 s analysis window and trims the given input signal for further analysis [35]. The trimmed core motion segment is processed to automatically identify the most relevant joint and limb parts for making predictions by using the AGCN model [34]. The AGCN model extracts feature representation by treating a given core motion sequence of 60 markers as a graphical model representing a human skeleton, where each node is marker position (joint) and the edge is connectivity between markers (bone) of ongoing kinematic sequence. The AGCN automatically learns the most important joint and bone motions across all samples (domain-dependent attention weights) and specific to given samples (input-specific attention weights) for predicting medication state, FOG score, and MDS-UPDRS-III total score, excluding FOG score.

#### 2.2.2. Trimming Core Motion Segment

Adaptive Trimming (AT) enables the model to identify core motion segments and to flexibly trim the signal that is most useful for specific prediction tasks of interest. From a previous study, AT was very effective at detecting gym exercise classification task [35]. In this work, the AT is fully trained with a given kinematics dataset to predict the start and end time of the core motion segment from a 4 s analysis window, $X\in R^{N\times C\times T}$.
(1)$$\begin{array}{cc} c & =sigmoid(F^{center}\left(F_{at}\left(X\right)\right)) \end{array}$$
(2)$$\begin{array}{cc} w & =\exp(F^{width}\left(F_{at}\left(X\right)\right)) \end{array}$$

$F_{at}$ is a four-layer convolutional network for extracting feature to predict core motion locations. $F^{center}$ and $F^{width}$ are two-layer fully connected models to predict center location, 0<c<1 and width of core motion segment, 0<w<1, which are further processed to derive start, *s*, and end, *e*, indices of given window, where 0<s,e<T.
(3)$$\begin{array}{cc} s & =T\times sigmoid(c-\frac{w}{2}) \end{array}$$
(4)$$\begin{array}{cc} e & =T\times sigmoid(c+\frac{w}{2}) \end{array}$$
(5)$$\begin{array}{cc} X^{C} & =X\left[s:e\right]=F_{crop}\left(X,s,e\right) \end{array}$$
(6)$$\begin{array}{cc} \; & =F_{sampler}\left(F_{grid\_gen}\left(X\right),s,e\right) \end{array}$$

The cropping operation adapts grid generator, $F_{grid\_gen}$, and sampler, $F_{sampler}$ that is used in spatial transformer network (STN) [38] to learn differentiable geometric manipulator function for cropping 2D images for the most salient object in the scene for image recognition. For AT, $F_{grid\_gen}$ generates 1D temporal grid with detected start, *s*, and end, *e*, indices of core motion signals and the temporal segment, $X\left[s:e\right]\in R^{N\times C\times T^{\prime }}$ is sampled with $F_{sampler}$, where $T^{\prime }=e-s+1$. This cropping operation resembles an interpolation process, which makes the whole AT model differential that can be trained with gradient back-propagation operation.

#### 2.2.3. Adaptive Graph Convolution

The trimmed core motion segment is represented as temporal graphical sequence, G=(V,E), where the node set, $V=\{v_{ti}t=1,\cdots ,T^{\prime },i=1\cdots \}$, includes markers (joints) in a skeleton sequence. The edge set is composed of two subsets, in which the first edge subset is the intra-skeleton connectivity (limbs) $E_{S}=\left\{v_{ti}v_{tj}\left(i,j\right)\in H\right\}$, where *H* is the set of connected joints defined by motion capture system, and second edge subset is the inter-frame edges, which connect the same joints in consecutive frames $E_{F}=\left\{v_{ti}v_{(t+1)i}\right\}$.

Given a temporal-spatial graph representation of motion segment, we first encode spatial dimension by using an AGCN [34] with $K_{v}$ kernel size, which is defined as follows,
(7)$$f_{out}=\sum_{k}^{K_{v}}W_{k}f_{in}\left(A_{k}+B_{k}+C_{k}\right)$$
where, $f_{in}\in R^{C_{in}\times T^{\prime }\times N}$ and $f_{out}\in R^{C_{out}\times T^{\prime }\times N}$ are input and output feature map and $W_{k}\in R^{C_{out}\times C_{in}\times 1\times 1}$ is weight vector of the 1×1 convolution operation. $A_{k}=\Lambda^{-\frac{1}{2}}\left({\bar{A}}_{k}\right)\Lambda_{k}^{-\frac{1}{2}}$ is a normalized N×N adjacency matrix of defined skeleton structure from our motion capture system, where ${\bar{A}}_{k}$ is a binary N×N adjacency matrix indicating the connectivity between the joints and $\Lambda_{k}^{ii}=\sum_{j}\left({\bar{A}}_{k}^{ij}\right)+\alpha$ is the normalized diagonal matrix using α=0.001 to avoid empty rows.

The attention maps of each node (joint) and edge (limb) are encoded in $B_{k}$ and $C_{k}$, which are learned fully data-driven manner. $B_{k}\in R^{N\times N}$ is an attention graph that encodes the underlining node and limb importance considering the entire samples of the task domain. $B_{k}$ is fixed once the parameters are trained and used for the inference. $C_{k}$ is an input-dependent attention graph to determine the strength of the connection between any two nodes in a given input graph sequence. Specifically, we applied embedded Gaussian Affinity [39] to calculate self-similarity between two nodes, $v_{i}$ and $v_{j}$ in a given input feature map, $f_{in}$.
(8)$$C_{k}^{ij}=f\left(v_{i},vj\right)=\frac{e^{\theta_{k}{\left(v_{i}\right)}^{T}\phi_{k}\left(v_{j}\right)}}{\sum_{j=1}^{N}e^{\theta_{k}{\left(v_{i}\right)}^{T}\left(\phi_{k}\left(v_{j}\right)\right)}}$$
Compared to $A_{k}$ and $B_{k}$, $C_{k}$ can flexibly attend to more important joint and limb motions according to changing inputs at inference time. Combining $A_{k}$ (predefined skeletal connectivity), $B_{k}$ (domain-specific connectivity), and $C_{k}$ (input-specific connectivity) helps the model to fully adjust the graphical structure of the input sample to only focus on the motion signals that are useful for jointly predicting medication state, FOG score, and MDS-UPDRS-III total score excluding FOG score. Additionally, we did not restrict the learned $B_{k}$ and $C_{k}$ to be left and right body symmetric to take into account the potential for asymmetric symptoms [40,41].

To further encode temporal dimension, $K_{t}\times 1$ temporal convolution is applied to spatial feature, $f_{out}$, extracted from the above mentioned attention-based graph convolution model, thereby, deriving spatial-temporal graphical representation, $f_{out}^{ST}=conv_{K_{t}\times 1}\left(f_{out}\right)$. In this study, we use a four-layer temporal-spatial graphical convolutional network (TGCN) with 64 feature maps to encode core motion in the given 4 s analysis window. Temporal Average Pooling (TAP) [42] is applied to the output of the last layer to summarize the feature across the temporal axis.

#### 2.2.4. Multi-Task Prediction

The feature representation from the last TGCN layer is used to simultaneously predict medication state, FOG score, and MDS-UPDRS-III total score excluding FOG score. (i) Medication state is a binary classification problem, either OFF or ON state. The feature representation is processed with two-layer fully connected model and a two-way softmax classifier, which is trained with binary cross-entropy loss. (ii) FOG score has 5 levels, from 0 (absent) to 4 (severe) FOG. For FOG score prediction, the feature representation is processed with two-layer fully connected model and five-way softmax classifier, which is trained with multi-class cross-entropy loss. (iii) MDS-UPDRS-III total score excluding FOG score is a positive integer ranging between 0 and 120. Before the model training, we apply Z-score normalization to marginalize the impact of outliers to bias the model prediction behaviors. To additionally consider the non-Gaussian distribution of the MDS-UPDRS-III total score excluding FOG score, we processed the feature representation with Gaussian Mixture Model (GMM) regression model [43].
(9)$$p\left(yx\right)=\sum_{i=1}^{m}\alpha_{i}\left(x\right)N\left(\mu_{i}\left(x\right),\sigma_{i}^{2}\left(x\right)\right)$$
where $x\in R^{D}$ is feature representation from the last TGCN layer, $\alpha =softmax(f_{\alpha }\left(x\right))$ is mixing coefficients for Gaussian distributions and $\mu =f_{\mu }\left(x\right)$ and $\sigma =exp(f_{\sigma }\left(x\right))$ are mean and standard deviation of each Gaussian distribution. For projection functions, $f_{\alpha },f_{\mu },f_{\sigma }$, two-layer fully connected models were used. In our experiment, the naive regression with a two-layer fully connected model and mean square error having a single Gaussian distribution assumption did not converge when training.

## 3. Experiment Setting

### 3.1. Model Hyperparameter, Training and Evaluation

For choosing model hyperparameters, we have followed the hyperparameter suggested in Ordóñez and Roggen [30], which was sufficient to predict fine-grained and complex human activities using movement data. (*i*) *AT*: Temporal kernel size and feature map were 3×1 and 64, respectively, for all four layers of the temporal convolutional model, $F_{at}$. Max pooling with $\times \frac{1}{2}$ was used at each output layer for aggregating temporal dimension. For predicting center location and width size of core motion segment, $F^{center}$ and $F^{width}$, two-layer fully connected layer model was used with 128 units and ReLU activation function [44]. (*ii*) *AGCN*: Four layers of the temporal graphical convolutional model were used, and kernel sizes of $K_{v}=3$ and $K_{t}=5$, respectively, for graphical and temporal convolution. Across all layers and convolutions, we used 64 feature maps, ReLU activation function, and max pooling with $\times \frac{1}{2}$ to aggregate along the temporal dimension. (*iii*) *Multi-task Prediction*: For two-layer fully connected models to predict medication state, FOG score, and GMM regression parameters, we used 256 and 128 units with ReLU activation function.

For the training model, we used a learning rate fixed at $1\times 10^{-3}$ with Adam optimizer and used a batch size of 16. Model training was stopped when no decrease in loss is observed from the validation set, which model is also used for evaluating the test set.

For evaluating the proposed method, we used 10-fold cross-validation. At each fold, 50%, 20%, and 30% of the dataset was used for the training, validation, and testing sets, respectively. We avoided placing adjacent analysis windows in different folds to avoid pairwise similarity biasing the cross-validation results [45].

To understand the potential use case for real-time analysis, we also compare the inference time between each baseline model, especially deep learning models. Based on standard benchmarking practices, we report an average of 1000 runs for a batch size of 16 using a NVIDIA QUADRO RTX 6000 with 24 GB RAM memory size.

### 3.2. Performance Metrics

For performance metrics, we used binary F1 score and mean F1 score for medication state and FOG score prediction, respectively, which is widely used for evaluating prediction performance in the presence of label imbalance. As shown in Table 2, most participants had FOG scores ≤2 for both OFF and ON medication states. The mean F1 score is an average of per-class F1 score, which is the harmonic mean of precision and recall of each class.
(10)$$\begin{array}{cc} Precision^{c} & =\frac{TP^{c}}{TP^{c}+FP^{c}} \end{array}$$
(11)$$\begin{array}{cc} Recall^{c} & =\frac{TP^{c}}{TP^{c}+FN^{c}} \end{array}$$
(12)$$\begin{array}{cc} F1\;score^{c} & =2\times \frac{Precision^{c}\times Recall^{c}}{Precision^{c}+Recall^{c}} \end{array}$$
(13)$$\begin{array}{cc} Mean\;F1\;score & =\frac{1}{C}\sum_{c}^{C}F1\;score^{c} \end{array}$$
where *C* is the number of classes and C=5 for FOG Score classification. For a class *c*, $TP^{c}$ is a true positive that represents the total of successfully classified class windows, $FP^{c}$ is a false positive that represents the total misclassified class windows, and $FN^{c}$ is a false negative that represents the total misclassified non-class windows.

For evaluating the regression performance for MDS-UPDRS-III total score excluding FOG score, we used root mean square error (RMSE).

### 3.3. Comparison with Baseline Models

We compared the proposed model to: (i) shallow models with hand-crafted features, and (ii) deep learning models including convolutional networks and graphical convolutional networks. We compared classifier performance across models using 95% Wilson score confidence intervals [46] for Medication State and FOG Score and using standard normal approximation based 95 % confidence intervals for MDS-UPDRS-III. For an imbalanced dataset, such as ours, Wilson score is widely used to calculate confidence intervals, as discussed in Appendix A.

*Shallow Baseline Models*. The first baseline models we considered were shallow models, such as Random Forest (RF) and Support Vector Machine (SVM) with radial basis function (RBF) kernel, with FOG-related hand-crafted features. Following previous work [13,47,48], we extracted various time, frequency, and distribution features, including freezing index [49], variance, sample entropy [50], central frequency, dominant frequency, and wavelet mean [51] features from the acceleration signals at multiple on-body locations. We used second-order Savitsky-Golay differentiation to derive acceleration traces from joint marker kinematics.

To investigate whether the lateralization of parkinsonian symptoms would impact model performance, We iterated RF and SVM models using markers from the left side of the body only (RF-L, SVM-L) and using markers from both sides (RF-LR, SVM-LR). We focused on lower body parts and independently trained RF and SVM for each task separately, following previous work [13,47].

*Deep Baseline Models.* We also compared the proposed model to several deep learning models for processing human skeleton time-series, including Temporal convolutional network (TCN) [52], Graphical convolutional network (GCN) [53], GCN with attention model (AGCN) [34], and AGCN with Adaptive Trimming (AT+AGCN). We used identical hyperparameters for model architecture and training wherever possible in order to make the fairest possible comparisons between deep learning models. All models were 4-layer with 64 feature maps and $\times \frac{1}{2}$ max pooling. For deep learning models, and for the classification and regression, we used two-layer fully connected layer with 256 and 128 units with ReLU activation functions.

### 3.4. Comparison with Single-Task Prediction

Since the proposed model is constrained to learn features relevant to three simultaneous prediction tasks, we reasoned that the identified features might be sub-optimal for single task prediction, leading to decreased performance. Therefore, we re-trained the deep learning models (with the exception of AT+AGCN+GMM) on the FOG score prediction task only and assessed changes in performance. We did not include the AT+AGCN+GMM model in this analysis as without the MDS-UPDRS-III prediction task it is identical to the AT+AGCN model.

### 3.5. Model Interpretability

We considered it critical to assess the clinical relevance of features identified by the model as relevant to medication state, FOG score, or total MDS-UPDRS-III score. These included individual kinematic markers (often referred to as “joints” in the computer vision literature) and segments (“limbs”) with high attention scores, and kinematic marker trajectories with high relevance to particular labels.

To derive overall model attention to individual segments or limbs, we aggregated attention maps across all samples in the dataset by averaging the learned attention maps and graphical structure over all *M* samples:(14)$$\begin{array}{c} E_{A}=\frac{1}{M\times K}\sum_{i}^{M}\sum_{k}^{K_{v}}\left(A_{k}+B_{k}+C_{k}^{i}\right)\in R^{N\times N} \end{array}$$
where $A_{k}$, $B_{k}$, and $C_{k}\left(x_{i}\right)$ are the normalized N×N adjacency matrix of predefined skeleton structure, the domain-wise N×N attention map, and the input-dependent N×N attention map at each kernel, respectively. The attention weights for joints and segments are then defined as the diagonal components $E_{A}^{jj}$ and off-diagonal components $E_{A}^{ij,j\neq i}$ of $E_{A}$, respectively.

To identify individual kinematic marker trajectories and core motion segments with high relevance to particular labels, we visualized individual analysis windows and core motion segments that the model predicted with high confidence, as measured by the entropy of the class prediction distribution. We visualized these data and discussed the interpretation with clinician experts within our project team, within the movement disorders group at our center, and at a regional forum in the Atlanta area hosted by the study sponsor in order to assess whether the identified features were consistent with the features that movement disorders specialists are trained to identify as characteristics of freezing.

### 3.6. Model Performance Furthermore, Potential Bias

After evaluating the proposed model against other candidate models, we assessed the potential for bias in model performance associated with participant demographics. After computing individual F1 score for each participant, we compared model performance across age and sex with linear models. Linear models used FOG study group (PD-FOG, PD-NoFOG, PP-FOG), dichotomized age, and sex as predictors of individual F1 score. Statistical significance was assessed with Wald tests at *p* = 0.05.

## 4. Results

### 4.1. Overall Model Performance

Here, we report the overall prediction performance of the proposed model (*AT*+*AGCN*+*GMM* in Table 3), compared with the performance of baseline models for predicting medication state, FOG score, and MDS-UPDRS-III total score excluding FOG item. In general, the proposed model’s performance was very high for both classification and regression tasks: Medication State, 97.6% cross-validated F1 score; FOG Score, 96.8% cross-validated F1 score; and MDS-UPDRS-III, 2.7 point RMSE, which is within the minimal clinically-important difference [54] for the instrument. In particular, the addition of the GMM regression component—which learns non-Gaussian distributions flexibly—to the second-best performing model architecture (AT+AGCN) significantly improved MDS-UPDRS-III performance. Performance of all models is summarized numerically in Table 3.

The prediction performance of the proposed model on MDS-UPDRS-III score excluding FOG item is shown in Figure 3. The overall RMSE was 2.7±0.4 points. As expected, overall, ON medication sessions have lower and OFF medication sessions have higher MDS-UPDRS-III total scores, as indicated by the higher prevalence of red points to the left of the plot and the higher prevalence of blue points to the right of the plot. We noted that the model tended to overestimate lower scores and under-estimate higher scores, as indicated by datapoints in the upper left and lower right.

### 4.2. Comparison to Baseline Models

For comparison to the previous state-of-the-art models in FOG analysis, we started analysis with shallow models (RF and SVM) using only lower body parts. We tested the use of left only and both left and right lower body parts (RF-LR and SVM-LR). Using both sides of body significantly improved performance on all three tasks; increasing F1 score by 12% and 33% for Medication State and FOG Score, respectively, and decreasing MDS-UPDRS-III RMSE by 10%. Among the shallow models (RF-LR and SVM-LR), RF significantly outperformed SVM, increasing F1 score by 7% and 13% for Medication State and FOG Score, respectively, and decreasing MDS-UPDRS-III RMSE by 9%, presumably due to its ability to learn non-linear decision boundaries.

Deep learning models also substantially outperformed shallow ML models, providing evidence that learning FOG representations that capture complex patterns may be more effective than using existing hand-crafted FOG features. The same conclusion was reached in a previous study using wait-worn inertial measurement units (IMUs) and standard deep learning models for analyzing FOG gait [55]. Deep learning models also outperform handcrafted features with various machine learning applications in human activity recognition [30,31]. Compared to the best performing shallow model (RF-LR), the TCN model, which mainly captures temporal patterns of each joint movement sequence, improved F1 score by 33% and 24% for Medication State and FOG Score, respectively, and decreased MDS-UPDRS-III RMSE by 43%.

Among the deep learning models, we also noted significant performance improvements in F1 scores among graph-based models vs. the more traditional TCN, as graph-based models can additionally capture positional relations between joints with a graphical data structure defined as a human skeleton. The simplest graph-based model significantly outperformed the TCN on all three tasks (4%, 9%, and 11% improvements on medication state, FOG score, and MDS-UPDRS-III, respectively). Further significant improvements were noted with the addition of attention mechanisms which enable the model to adaptively concentrate its representation powers for the most relevant joint depending on the given input (4%, 2%, and 8%). The additions of adaptive trimming and the Gaussian mixture model prediction did not significantly improve F1 scores, but significantly reduced MDS-UPDRS-III RMSE (4% and 23%, respectively). We speculate that the flexibility of the GMM model stabilized the gradient backpropagated from the regression branch to help find a more effective feature representation for all tasks. Runtime increased with model complexity, as expected. Nevertheless, the proposed model, which is the most complex, showed a sub-second inference time, enough for real-time analysis.

### 4.3. Comparison to Single-Task Prediction

All four deep learning models tested showed significantly improved performance on FOG score prediction when trained on the multi-task problem (medication state, FOG score, and MDS-UPDRS-III) rather the single task problem (FOG Score only). When the models were trained on the single task problem, the TCN, GCN, AGCN, AT+AGCN demonstrated F1 scores of 0.825±0.016, 0.892±0.033, 0.903±0.047, and 0.925±0.012, respectively, a 3.8% *decrease* in performance on average compared to the multi-task problem. We speculate that additional information provided to the models by predicting medication states and MDS-UPDRS-III total scores helped to learn representations that are more targeted and personalized to discriminate detailed differences in FOG phenotypes in varying PD conditions, which eventually helped improve overall FOG score classification performance.

### 4.4. Model Interpretability

#### 4.4.1. Most Relevant Joints and Limbs

We visualized markers and segments with the top ten largest attention weights to assess which body parts were most salient to the prediction task (Figure 4). Attention weights were concentrated on the head, chest, waist, hands, and (particularly left) legs. We suggest that the attention paid to markers on essentially all body segments reflects the fact that FOG is a full-body phenomenon, and suggests that the model may be attending to en-bloc turns [56]—which tend to be maintained across medication states [57]—or other elements of impaired intersegmental coordination. We noted that in particular, the model attended closely to segments on the left foot, which had been suggested previously by a clinical expert on our team as relevant to FOG in this testing condition, which requires left turns. Interestingly, the model also attended to the fingers and elbows. Although these body parts are not typically attended to during clinical FOG examination, patients with FOG can also freeze during upper limb movements [58], leaving open the possibility that the model was attending to hand movements characteristic of freezing.

#### 4.4.2. Most Relevant Motion Segments

We also visualized patterns of left heel movement that were predicted as relevant to particular medication states and FOG scores with high confidence. Figure 5 shows the detected core motion segments of left heel movements from the adaptive trimming model during the timed-up-and-go trials of patients with ON and OFF medication states and different FOG scores. In general, the adaptive trimming model automatically captured approximately a single step cycle within each 4 s analysis window with movement patterns especially related to FOG.

The identified kinematic associated with detected core segments were highly consistent with the features that movement disorders specialists are trained to identify as characteristics of freezing. This analysis shows that the model focuses on periods with regular gait activity for epochs corresponding to FOG scores of 0 and 1, and periods of interrupted gait activity or pure akinesia for epochs with higher scores. For the samples with a FOG score of 0 (first column), the model considered normal stepping gait and used a walking cycle motion for making predictions. For the samples with a FOG score of 1 (second column), the models detected decreasing step length from the motion automatically. As the FOG score became higher, the model tended to detect more FOG-related gait motions. For the samples with a FOG score of 2 (third column), the model detected onset of gait signal related to festination (tendency to speed up in parallel with a loss of normal amplitude of repetitive movements) [59]. For the samples with FOG score of 3 and 4 (fourth and fifth columns), the model detected freezing gait, akinesia, and trembling signals as core motion signal that is relevant for predicting FOG scores.

### 4.5. Classifier Performance Furthermore, Potential Bias

After computing individual F1 score for each participant, we compared model performance across study and demographic groups to assess potential bias. Linear models found no significant differences in F1 score across study groups or sex, but found significantly decreased performance (reduction in F1 score of 17%, *p* < 0.01) among older participants (age ≥ 69 years) compared to younger participants.

## 5. Discussion and Conclusions

In our experiment, we designed a deep neural network model to simultaneously predict levodopa medication state (ON/OFF), FOG score (0–4), and MDS-UPDRS-III total score (less FOG score) from full-body kinematics data of 57 patients, including 5 patients with atypical parkinsonism, assessed with TUG tests in the off and on medication state. As compared to formal clinical assessments by a movement disorders specialist, our AGCN model classified levodopa medication state and FOG score with 96.4% and 96.2% F1 scores, respectively, and regressed MDS-UPDRS-III total score with root mean square error (RMSE) of 2.7 points.

To the best of our knowledge, this is the first work that applies an interpretable deep learning model with full-body kinematics for classifying FOG. This model detects time segments having characteristic movements of FOG during walking, e.g., small shuffling steps, akinesia, and tremulousness. Additional findings demonstrated that FOG is not limited to the lower extremity, and also significantly involves movements in the upper body, further supporting that FOG requires phenotyping using whole-body kinematics. Findings from our analysis may lead to novel hypotheses to define more granular FOG phenotypes, or potentially to technologies that enable continuous monitoring of FOG severity in order to test new therapies with improved precision.

Overall, while the current study uses 3D kinematic data, we believe that the underlying approach will generalize to motion estimates obtained through pose estimation or body-worn sensors, enabling future applications in clinical and home settings with 2D video. The patterns of body motion recorded here result from fundamental principles of physics and biomechanics, which are likely to hold regardless of the method used to measure motion. For example, the laws of motion and principles of energy conservation apply regardless of whether motion is measured using 3D kinematic data, pose estimation, or body-worn sensors. This is likely why it is feasible to estimate virtual IMU signals from video data [35]. The proposed model is expected to be also useful for discovering novel phenotypes for other types of movement disorders, such as, Cerebral Palsy [60], Ataxia [61], or Essential Tremor [62], which conventionally use hand-crafted features for the analysis.

The study has three main limitations. First, we did not attempt to identify freezing of gait (FOG) at the millisecond level, which would be necessary for use in assistive technology. Second, we did not attempt to measure FOG severity as a continuous outcome, which could increase precision in clinical trials. Finally, the study sample was predominantly white and had fewer females than would be representative of the Parkinson’s disease (PD) population [63], so the generalizability of the results to the entire PD population may be limited.

One primary contribution of this work is the application of deep learning to the problem of scoring FOG, which has primarily been examined with hand-crafted and engineered features such as spectral power in a prespecified “freeze band” [16] calculated from a prespecified set of body segments. Indeed, despite the typical notion that FOG is an interruption of walking—leg movements—our results indicate that scoring FOG with high accuracy may require attention to body parts across the body, including the hands and head. We believe that adopting a data-driven approach with explainable deep learning models represents an important way forward in modeling kinematics from walking and turning motions of parkinsonian patients.

We hope that using deep learning to discover data-driven kinematic features will lead to the development of a more fine-grained and objective FOG severity scales, which could provide valuable information to clinicians and researchers, help to improve diagnosis, treatment, and overall management of FOG (cf. [10]).

## Figures and Tables

**Figure 1 sensors-23-01766-f001:**
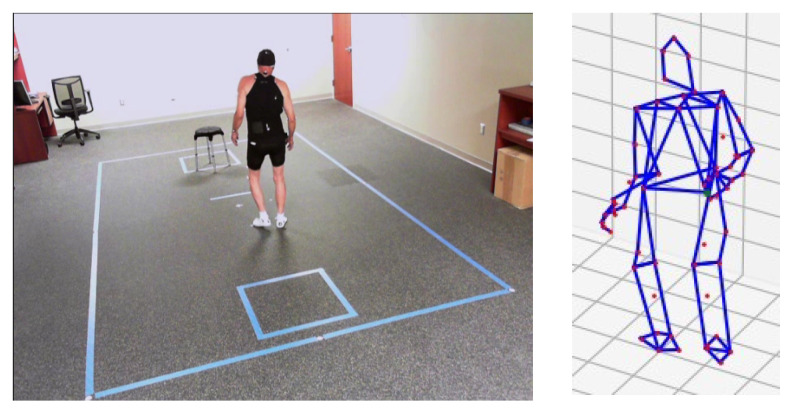
Motion capture recording during timed-up-and-go testing. **Left**: clinical motion capture laboratory. **Right**: example of kinematic marker data. Participants were instructed to rise from the stool, walk to the taped box, and return three times during each test.

**Figure 2 sensors-23-01766-f002:**
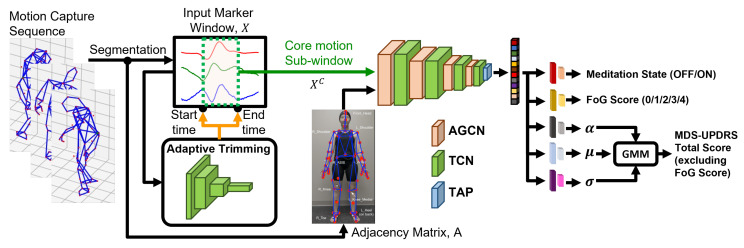
Overall model architecture. The recorded motion capture sequence is segmented into 4 s analysis windows, which are first processed with the Adaptive Trimming (AT) model. AT model, which uses a 4-layer temporal convolutional network (TCN), predicts the start and end index of the core motion segment that is most relevant for the prediction task. The core motion segment is processed with a 4-layer adaptive temporal-spatial graphical convolutional network (AGCN), which automatically learns the attention map for the most relevant joint and limb motion for the prediction task. The feature representation from the final layer of AGCN is processed with temporal average pooling (TAP) to summarize temporal information, which is then used to predict medication state, FOG score, and MDS-UPDRS-III total score (excluding FOG score) at the same time. Specifically for regressing the MDS-UPDRS-III total score (excluding FOG score), Gaussian Mixture Model (GMM)-based regressor is used to take account of the non-Gaussian distribution of the target values.

**Figure 3 sensors-23-01766-f003:**
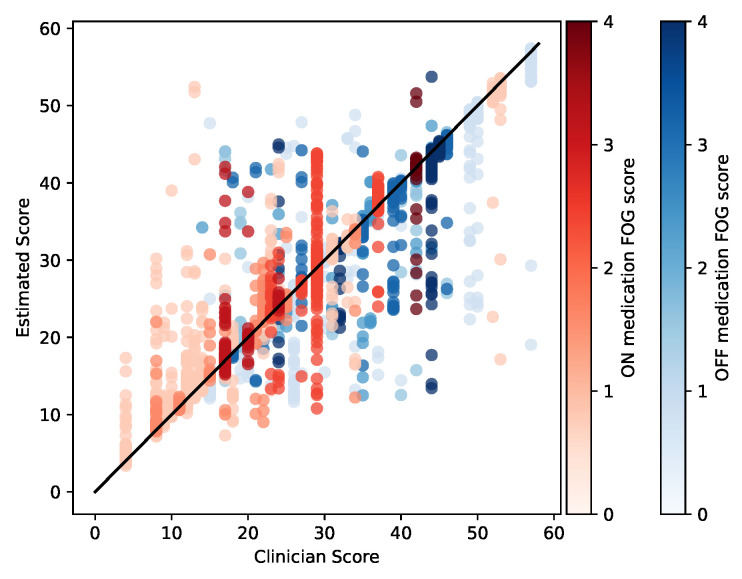
Scatter plot comparing clinician-rated versus model-estimated MDS-UPDRS-III total score, excluding the FOG item (3.11). Unity line is shown for reference. Each dot represents a single 4 s analysis window. Colors are used to represent the FOG item scores corresponding to each analysis window, with darker colors indicating more severe FOG in the OFF (blue) and ON (red) medication states.

**Figure 4 sensors-23-01766-f004:**
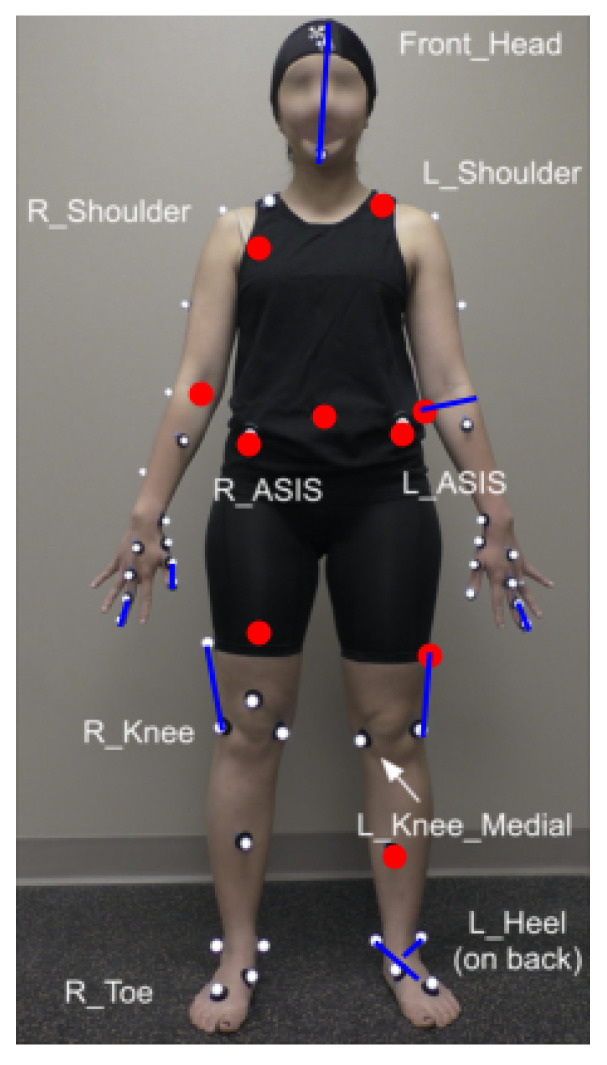
Kinematic markers (referred to as “joints” in pose estimation literature, red) and segments (referred to as “limbs” in pose estimation literature, blue) with the top 10 attention weights in the prediction tasks.

**Figure 5 sensors-23-01766-f005:**
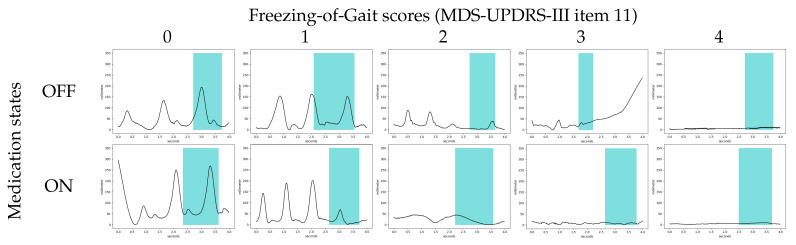
Examples of detected core motion segments of left heel motions from the adaptive trimming model for different medication states and FOG scores. The horizontal and vertical axes of plots are in seconds and millimeters, respectively, with the vertical axis indicating the height of the left heel marker above the laboratory floor. All plots depict four seconds of recorded movement. Core motion segments detected by the adaptive trimming model are depicted in blue.

**Table 1 sensors-23-01766-t001:** Clinical and demographic features of study participants.

	PD-FOG	PD-NoFOG	PP-FOG
N	35	17	5
Age, y	69±7	67±12	66±6
Sex, M/F	30/5	11/6	2/3
Disease duration, y	10.5±6.7	6.0±3.6	6.0±3.3
LED, mg	1429±673	833±303	1258±640
MDS-UPDRS-III (OFF)	34.0±10.6	30.8±13.2	39.4±7.8
MDS-UPDRS-III (ON)	20.7±8.7	18.4±14.5	31.6±9.0
NFOG-Q	20.1±4.9	0.0±0.0	17.8±7.5

**Table 2 sensors-23-01766-t002:** Summary of timed-up-and-go testing sessions stratified by medication state and FOG score.

Medication	FOG Score				
State	0	1	2	3	4
OFF	21	15	9	8	7
ON	38	11	7	3	1

**Table 3 sensors-23-01766-t003:** Prediction performance of the proposed model (AT+AGCN+GMM) and comparison to baseline models. We also compare the run time of each deep learning model used in the experiment. Performance metrics are presented as mean ± 95% confidence interval. Deep learning models are indicated by italics. Abbreviations are described in text. ${}^{a}$ Total score with FOG item (3.11) subtracted. ${}^{†}$
*p* < 0.05, improvement in RF vs. SVM. ${}^{‡}$
*p* < 0.05, improvement in -LR vs. -L. * *p* < 0.05, improvement in deep learning models vs. preceding row. ${}^{§}$
*p* < 0.05, improvement in multi-task vs. single task prediction.

	Medication State	FOG Score	MDS-UPDRS-III ${}^{a}$	Inference Time
Model	(F1)	(F1)	(RMSE)	(Seconds)
SVM-L	0.540±0.016	0.429±0.026	9.346±0.138	-
RF-L	$0.594\pm 0.012{\;}^{†}$	$0.553\pm 0.038{\;}^{†}$	9.189±0.301	-
SVM-LR	$0.616\pm 0.017{\;}^{‡}$	$0.608\pm 0.031{\;}^{‡}$	$8.714\pm 0.101{\;}^{‡}$	-
RF-LR	$0.657\pm 0.019{\;}^{†,‡}$	$0.684\pm 0.040{\;}^{†,‡}$	$7.918\pm 0.427{\;}^{†,‡}$	-
*TCN* [52]	0.875±0.017 *	0.851±0.020 *${}^{,§}$	4.551±0.276 *	0.0055
*GCN* [53]	0.913±0.015 *	0.929±0.021 *${}^{,§}$	4.023±0.373 *	0.0433
*AGCN* [34]	0.949±0.010 *	0.948±0.018 *${}^{,§}$	3.703±0.300 *	0.0435
*AT+AGCN*	0.955±0.021	$0.955\pm 0.026{\;}^{§}$	3.555±0.394 *	0.0469
*AT+AGCN+GMM*	0.975±0.018	0.967±0.022	2.753±0.440 *	0.0471

## Data Availability

The deidentified raw data and code for supporting the evidences in this work will be made available by the corresponding author upon reasonable request.

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
