# Peer review of "An Explainable Spatial-Temporal Graphical Convolutional Network to Score Freezing of Gait in Parkinsonian Patients"

_sensors, 2023, doi:10.3390/s23041766_

Round 1

Reviewer 1 Report

This manuscript put forward a novel deep learning method for scoring freezing of gait in parkinsonian patients, where an explainable spatial-temporal graphical convolutional network was developed for the task of interest. The performance of the proposed method was validated using experimental data, with high accuracy. The outcome of this research can be considered as a potential solution to discover novel kinematic biomarkers for FOG that can be used for 16 hypothesis generation and potentially as clinical trial outcome measures. Overall, this is a very interesting paper, with well-organised structure. It just needs a minor revision before being accepted for publication. Details of comments are shown below.

1.       The main innovation and contribution of this research should be clearly clarified in abstract and introduction.

2.       Broaden and update literature review on deep learning (such as convolutional networks) and its applications. E.g. Vision-based concrete crack detection using a hybrid framework considering noise effect.

3.       There are several hyperparameters that directly influence the generalisation ability of the proposed network. How did the authors find the optimal values to achieve the best prediction result in this research?

4.       Running time should also be considered as an evaluation metric for its practical application.

5.       More future research should be included in conclusion part.

Reviewer 2 Report

This paper proposess a method to measure Freeze of Gait (FOG) in patients with Parkinson disesase by training interpretable DLL models with multitask learning. The authors state that is this the first paper with an application of an interpretable DNN with full kinematics that classifies FOG.

The paper is very interesting and very well written. The methodology is explicitly described and the approach can be perceived by a wider audience. The authors compare the results of their approach with other ML and DL applications and their method performs better. Besides the issue of algorithm performance, I believe the approach is quite interesting to read. Jst two points to be checked/answered:

-table 3 that compares results: ML algortihms achieve low F1 scores, quite lower than DL approaches. Is that something common in this type of data? Were there an issues of overfitting?

-line 320 please check for the text exceeding the text body
